# Exploring dairy heifers' consistency in social motivation in the absence or presence of conspecifics

Sarah Kappel, Emeline Nogues , Daniel M. Weary, Marina A. G. von Keyserlingk *

Animal Welfare Program, Faculty of Land and Food Systems, The University of British Columbia, Vancouver, British Columbia, Canada

☯ These authors contributed equally to this work.
* marina.vonkeyserlingk@ubc.ca

## Abstract

Cattle are motivated to maintain social contact, but individual preference for social proximity (i.e., 'sociability') varies among individuals. Although personality traits, like sociability, are generally considered to be consistent across context and time, different social environments may elicit different behavioral responses in individuals. We tested individual differences in social motivation with and without the presence of conspecifics, and compared responses within and among different social contexts. Specifically, Holstein heifers (n = 36) were exposed to standardized social isolation tests (novel arena, novel object, runway test; each tested twice, 14 days apart) and a novel social-feed trade-off paradigm. In addition, heifers were subjected to a distribution test in which groups of three animals could freely move between two feed troughs. At one trough, heifers were offered 150 g of grain every two minutes (high side) and every four minutes at the other trough (low side). We expected animals to disperse proportionally to resource availability, corresponding with the Ideal Free Distribution theory (IFD), and that deviations from IFD would reveal individual differences in motivation to be with peers. We found no consistency in sociability measures derived from behaviors in the absence (vocalization, runway latency) and presence (feeding time, social-feed trade-off score) of conspecifics. Moreover, behaviors showed low repeatability within the same social contexts. We conclude that individual differences in sociability are likely context-dependent. We suggest that sociability might not be a single 'trait', but rather a 'behavioral consequence' resulting from the combined effects of internal characteristics (e.g., other personality traits such as fearfulness) and external factors (e.g., testing environment) at the time of evaluation.

**Data availability statement:** https://doi.org/10.5683/SP3/AGBUKH.

**Funding:** This work was funded by a Canada's Natural Sciences and Engineering Research Council (NSERC) Discovery Grant (#F20-04644) awarded to MvK. The funder played no role in any aspect of the research.

**Competing interests:** no authors have competing interests.

## Introduction

Cattle are motivated to maintain proximity to conspecifics [1,2], but the degree of social motivation can vary among individuals [3]. Individual differences consistent across time and context are referred to as 'personality traits' [4–6]. In dairy cattle, the personality trait 'sociability' is commonly assessed using a runway test [3]; measured as latency to reunite with conspecifics when a test animal is socially isolated and then given the choice to return to its group. In this case, 'sociability' is assessed using a measure that relates to how strongly motivated the animal is to regain proximity to peers. Social separation is fear-eliciting in cattle [7] and, as such, induces stress [8]. Under natural conditions, cattle do not seek social isolation, except during calving [9,10]. Individual responses to social separation may be confounded by other factors influencing social behavior, such as individual differences in coping style and fearfulness [11].

Behavioral responses during social isolation tests, such as the novel arena and novel object test, are often used to assess personality traits (e.g., fearfulness; [12]) in social animals, despite concerns about the ecological validity of these tests (see critical reviews by [7,13]). One limitation is the interpretation of specific personality traits observed from animal behavior (discussed in [14]). Some authors have used locomotion and vocalization behaviors during novel arena tests as an indicator of sociability [15], but others have used these same responses as an indicator of fearfulness (reviewed in [7]) or as a signal of separation distress [16]. Novel object tests are typically used to assess exploration and boldness, but responses may also relate to sociability, given that animals are tested in social isolation. Indeed, these responses may reflect more than one personality trait [14]. For instance, testing dairy cattle individually with novel objects in a familiar environment, Hasenpush et al. [17] identified 'sociability', 'exploration' and 'boldness' as three traits explaining variation in dairy cows' response to novel object tests.

Assessing the behavior of animals within their group may provide a more naturalistic assessment of sociability without the confounding effects of social isolation [18]. Proximity-based observations have been used as an indicator of sociability in intensively housed dairy cattle, with more sociable animals spending more time in proximity to peers (e.g., [3]). However, restricted spaces can limit individual expression of social preferences and social proximity is likely influenced by additional factors social dominance [19,20] and management factors such as competition for feed bunk access [21].

In free-ranging herbivores (sheep, cattle), 'sociability' has been assessed by measuring animals' willingness to move away from peers in exchange for feed; whereby, a lower level of social motivation is reflected in willingness to approach feed at greater distances to peers [22–25]. That is, more sociable sheep are less likely to move away from peers to obtain a grain reward compared to less sociable sheep [22,23,25]. Inter-individual variations in social-feed trade-off willingness have also been reported among beef cattle [24,26,27].

Differences in social motivation may help explain why free-ranging animals do not always distribute themselves among alternative foraging sites in such a way as to

maximize foraging gains, in conflict with the prediction of the Ideal Free Distribution model (IFD; [28]). The IFD predicts that, assuming that all animals have the same competitive abilities, 'ideal' knowledge of the relative quality of different food patches and are 'free' to move between resources to achieve the same gains [28], animal distribution is proportional to resource availability. However, conflicts between group members seeking proximity with conspecifics versus seeking to disperse to access more beneficial feeding locations may lead to deviations from IFD [29]. For instance, sheep show trade-offs between staying closer to conspecifics and grazing at preferred feeding locations at greater distances to peers [23]. This shows that social motivation is not a negligible cost when trying to understand the drivers of animals' interaction with their physical and social environment. To our knowledge, no research has applied the concept of IFD, or deviations thereof, in intensively housed dairy cattle.

Behaviors of social animals observed when tested alone might differ from their responses in the presence of conspecifics [30–32]. Some studies have reported low repeatability in behavioral responses used to assess sociability when cattle were tested in the absence and presence of conspecifics (e.g., [30,33]), suggesting that different testing contexts might reflect different motivations [32]. However, direct comparisons of contextual stability of individual variations in personality traits in cattle, including sociability, are still limited [30].

Understanding how preference for social proximity differs among dairy cattle and whether social motivation differs across contexts (i.e., in the presence versus in the absence of conspecifics) is important to provide more tailored management practices to meet each animal's needs regarding its social and physical environment [30,34]. Therefore, this study aimed to compare responses used to assess social motivation with and without the presence of conspecifics. Responses in social isolation were measured in a novel arena, a novel object and a runway test. To test social motivation in the presence of conspecifics, we used two test paradigms. In a social-feed trade-off test (adapted from [24]), we tested individuals' propensity to move away from peers in exchange for grain, anticipating that more sociable animals would be less likely to move away from conspecifics. To assess whether social spacing in a group feeding context reflects social motivation, we developed a distribution test. In this test, groups of three heifers could move freely among two separate feeding troughs, one providing twice as much grain (high side) as the other (low side). We predicted that, on average, each individual would spend 33% of their time feeding alone from the low side, conforming to the IFD model.

Corresponding with the previous findings indicating that behaviors assessed in cattle in response to isolation show consistency across time [30,34], we expected to observe high levels of agreement among responses to the novel arena, novel object, and runway tests. Likewise, we hypothesized that responses to the distribution test would be positively associated with the willingness to exchange feed for social proximity in the trade-off test. Finally, we explored the associations between sociability measures observed in the absence and presence of conspecifics.

## Methods

### Ethical approval

This study was conducted at The University of British Columbia Dairy Research and Education Centre (Agassiz, BC) under the approval of the university's Animal Care Committee (#A23-0090).

### Animals & housing

We enrolled 36 Holstein Friesian heifers (see S1 Appendix for sample size justification), with 21 of these animals having been used in another study, 6 months before this study in [35], which included short periods of social isolation, human handling, and exposure to enrichment items. Heifers were 6.41 ± 1.06 months old (mean ± SD; here and elsewhere) and weighed on average 230 ± 26.73 kg at the beginning of the study.

Animals were randomly assigned to one of three groups (of 12 heifers each), housed together for at least two months before the study. Groups were tested sequentially between March and May 2024. During testing, each group of heifers was housed in the same free-stall pen (approximately 4.8 m² of space per heifer) fitted with 13 deep-bedded sand stalls (2

x 0.9 m each). The feeding area was accessed through a feed barrier containing 15 self-lock head gates (35 cm center-to-center). Fresh feed was delivered daily between 8.30–9.30 am and pushed up every 2 h (except between 4.30 am and 11 am) by an automated feed pusher (DairyFeed F4800, GEA, Germany). Rations for the heifers were formulated according to the recommendations for dairy cattle [36]. Given the three-month duration of the study, there were differences in the forage component of the ration provided to the three study groups (group 1 and 3 received a total mixed ration (TMR) composed of straw, alfalfa hay, grass, and corn silage, group 2 received grass silage; feed provided *ad libitum* to all animals). In addition, each animal received ~1 kg grain (18% flaked calf starter, crude protein 18.0%, crude fiber 6.7%, crude fat 1.1%, phosphorous 0.65%, sodium 0.26%, vitamin A, D3, E 9011, 2007, 66 IU/kg, respectively). The grain was added on top of the *ad libitum* feed (i.e., 'top-dressed') daily between 9.30 and 10 am (approximately 2 to 2.5 h before testing), except on test days, when the top-dressed amount was reduced by 50% as animals received more grain during the social-feed trade-off and distribution tests (see test descriptions). A self-filling trough provided *ad libitum* access to water.

## Experimental design

Each heifer was subjected to five different behavioral tests, including a series of three standardized personality tests: novel arena, novel object, and runway test. We show the timing and order of each test in relation to the other tests in Fig 1. First, single animals were tested successively in novel arena and then novel object test. Between the two tests, each heifer was subjected to a trade-off test to measure their willingness to move away from conspecifics in reward for food. A second trade-off test trial was conducted after the novel object test. Due to time limitations, six randomly selected animals were tested on the same day, and the remaining six animals were tested the following day. The runway test was conducted individually on all heifers on a single day after completion of the novel arena, novel object and trade-off tests. All four behavior tests were conducted twice, with a two-week interval between test days. All individuals were tested in the same order as described above in the first and second repeats of the personality tests.

We also recorded the dispersal of three heifers among two feeding sites in a distribution test. Within each group of 12 heifers, animals were assigned randomly (see S2 Appendix) to a subgroup (4 within each group, 12 subgroups in total; subgroups remained the same for all distribution test sessions). Before conducting any behavior tests, we record three 24-h periods of home pen behavior to assess each individual's social rank within the group.

## Social isolation tests

The novel arena and novel object tests were conducted in the same arena (sawdust bedded area of 9.3 m x 5 m), allowing no visual but auditory contact to conspecifics. An unfamiliar object (novel object in first repeat: a round blue plastic

| Day -3 to -1 | Day 1 | Day 2 | Day 3 | Day 4 | Day 5-7 | Day 8-12 | Day 13 | Day 14 | Day 15 | Day 16 |
|---|---|---|---|---|---|---|---|---|---|---|
| Home pen Recordings | NAT SFT1 NOT SFT2 | NAT SFT1 NOT SFT2 | RWT | BREAK | DT Habituation | DT Testing | BREAK | NAT SFT1 NOT SFT2 | NAT SFT1 NOT SFT2 | RWT |
| (n=12 heifers) | (n=6 heifers) | (n=6 heifers) | (n=12 heifers) | | (n=12 heifers) | (n=12 heifers) | | (n=6 heifers) | (n=6 heifers) | (n=12 heifers) |
| | Personality test repeat 1 (PT1) | | | | | | | Personality test repeat 2 (PT2) | | |

**Fig 1. Timeline of behavioral tests conducted with each group of 12 heifers.** In the first repeat of personality tests (PT 1), each animal first underwent a novel arena test (NAT), then one social-feed trade-off test trial (SFT 1), followed by a novel object test (NOT) before a second SFT trial (SFT 2) was conducted. Half of the group was tested on day 1, the second half on day 2 due to time limitations. The next day, each heifer was tested in a runway test (RWT). After a one-day break, heifers were habituated to the distribution test (DT) for three days before being tested in the DT over five consecutive days. Following a one-day break, the personality test series was repeated (PT 2) in the same order of tests and animals as in the first repeat (PT 1). Before conducting any tests, animal behavior in the home pen was monitored for social dominance assessment on three consecutive days.

board (Ø 0.5 m); novel object in second repeat: a yellow bucket (0.4 m wide, 0.5 m long, 0.3 m deep), attached to the wall at 1 m height at approx. 6 m distance to the arena entrance. The novel arena and novel object tests lasted 10 and 5 minutes, respectively. Response behaviors recorded in these tests were video-recorded (GoPro HERO 9, USA). Two independent observers assessed the percentage of time each animal was active, standing, or exploring the arena and counted the frequency of vocalizations using BORIS ([37]; see S1 Table for details). For the novel object test, we additionally measured the latency to contact the novel object upon arena entry and the percentage of time spent with object exploration. Inter and intra-observer reliability for behavior durations using 20% of videos were in good agreement (intra correlation coefficient (icc) >0.84, see S3 Appendix). For the runway test, a single animal was gently moved into a holding pen at one end and three companion heifers were kept in another pen at the end of a runway (2 x 25 m). After 5 minutes in the holding pen, the gate was opened and the test animal's latency to re-establish proximity (within 5 m) to the group was recorded live by two independent observers who sat behind plywood walls on either side of the holding pen (icc = 1). The two observers also recorded vocalizations emitted by the test animal in the holding pen (icc = 0.92).

### Trade-off test

Our test design was based on a previous study in cattle [24], where an animal's tendency to leave its companions was inferred from the maximum distance animals moved away from companions to obtain grain from buckets placed at increasing distances to the group (up to 130 m). Due to space limitations, our test arena included four feed buckets (Ø 60 cm, depth 30 cm) within a U-shaped arena (9.3 m x 12 m, see Fig 2); visual contact between the test animal and companions (when present in the companion pen) was possible from bucket 1 and 2 but not from bucket 3 and 4. Grain was used as a food reward as it is a highly valued food source for cattle, and dairy cows will preferentially sort TMR for the

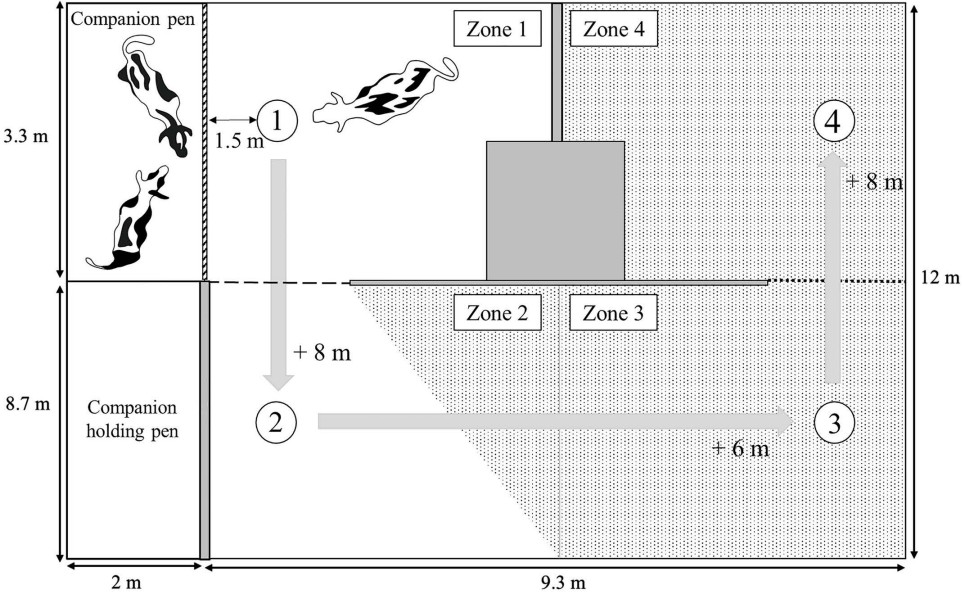

**Fig 2. Schema of the trade-off test setup.** Four buckets were placed at increasing distances from the companions. Buckets 1-3 contained 100 g, and bucket 4 contained 200 g of grain. Companions were visible from bucket 1, partially visible from bucket 2, and were not visible from buckets 3 and 4 (dotted test area). The dashed line between zones 1 and 2 indicates the threshold heifers had to pass within 5 minutes after starting to feed from bucket 1 to gain access to buckets 2-4. Crossing this threshold from zone 2 back into zone 1 terminated the test. Grey shaded areas indicate structural barriers within the test area, limiting visual contact with peers. Grey arrows indicate the anticipated order of the buckets that the animals would approach. All heifers were tested in two trials, one with conspecifics present in the companion pen and one without, by moving the conspecifics into the companion holding pen.

grain component [38]. Before testing, animals were habituated to feed from bucket 1 in groups of three, but not from the other bucket locations.

Animals could consume up to 1 kg/d during testing (100 g in bucket 1–3, 200 g in bucket 4; two trials per test day). Whether companions were present in the companion pen or not during the test was balanced for order, with half tested first with companions and half tested first without companions. The companions were part of the same subgroup of three heifers later tested in the distribution test. On trade-off test trials without social contact, the companions were moved to a holding pen (6.6 m²) adjacent to the companion pen, visually separating them from the test animal. During the test, each animal was given a maximum of 5 minutes to enter zone 2, otherwise, the test was terminated. After entering zone 2, the animal could freely explore between zones 2 and 4. The test ended when the animal re-entered zone 1 with all four feet, or after 10 minutes, whichever came first. One observer live-recorded the number of vocalizations emitted by the companion animals during both trade-off test trials (i.e., while visible and not visible to the test animal). Outcome measures (see Table 1) were derived from video analysis (inter-observer reliability, icc = 0.97). Test animal vocalizations were not recorded as they rarely vocalized.

## Distribution test

Three heifers were moved into the same test pen as in the trade-off test (without buckets) and offered grain at two feed troughs (see S1 Fig for test arena image); each trough providing ~1.5 m feeding space per animal when all three heifers fed from the same trough. The two troughs were separated by a plywood wall blocking visual contact. To move between troughs, heifers had to walk around the plywood barrier (~26 m walking distance between troughs). Following habituation (described in S4 Appendix), distribution test sessions were conducted with each subgroup of three heifers on five consecutive days. Grain was replenished at both troughs following two different intervals. Each trough was filled with 150 g of grain (see S5 Appendix for decision process on grain amount) every two (high side) or four minutes (low side; sides counter-balanced across subgroups). Once grain was delivered into the troughs, feed was only replenished if animals visited a trough (i.e., grain had been consumed) within the two or four-minute interval. This way, grain did not accumulate over time if not consumed. Our goal was to ensure that the difference in available grain provided in the two troughs was an incentive for the animals to move between feeding sites, but by design, we kept the difference in potential gain between sides to a minimum (300 g at the high side versus 150 g grain at the low side within a 4-minute interval). Theoretically, if animals dispersed so that two animals were feeding from the high side and one animal from the low side, all animals would receive the same amount of grain (i.e., following the Ideal Free Distribution where they distribute themselves proportionally to the availability of resources; [28]). However, if two animals fed from the low side and one from

**Table 1. Outcome measures recorded in the trade-off test.** Willingness to move away from the two companions kept in a pen adjacent to the test area in trade-off with grain was measured. Four buckets were placed at increasing distances (the first bucket at 1.5 m, the fourth bucket at 22 m distance) from the companions. Moving towards the buckets, the test animals' ability to maintain visual contact with their companions decreased and was completely obstructed at buckets 3 and 4.

| Recorded outcome measures | Description |
|---|---|
| *Latency to leave zone 1 after feeding from bucket 1* | The time when the animal stepped away from bucket 1 (i.e., consumed all the grain) to leave zone 1 by moving all four feet over the threshold line between zones 1 and 2 (see Fig 2). |
| *Latency to reach bucket 4 after leaving zone 1* | The time passed between the animal moving all four feet over the threshold line between zone 1 and 2 and starting to feed from bucket 4. |
| *Return latency from bucket 4* | The time from when the animal stopped feeding from bucket 4 until it re-entered zone 1. |
| *Time spent feeding from bucket (1–4)* | The time passed between the animal started feeding from a bucket until it stepped away from the same bucket for the first time. |

the high side, the latter animal could consume twice as much grain as the others. In this case, the grain at the low side would deplete faster and the animals should be motivated to switch troughs. In some cases, this motivation to switch may conflict with the motivation to remain with a peer, hence requiring a trade-off of social proximity for food. At the test session start, all three heifers were gently moved to the trough with the low side. Two handlers (not visible to the animals) positioned behind the troughs delivered 150 g of grain at each side in a synchronized manner while a third handler (with their back to the test arena for blinding) shook a bucket of grain for 10 seconds. After two minutes, the bucket was shaken again, and grain was simultaneously delivered into the trough according to the feeding schedule. No replenishment of grain occurred if none had been consumed (i.e., no heifers moved from the low to the high side before the replenishment at the high side was due). This sequence of shaking the bucket and replenishment was repeated every two minutes. Over the 22-minute test duration, a maximum of 1.65 kg and 0.9 kg of grain was delivered to the high (11-grain deliveries maximum) and low (6-grain deliveries maximum) side, respectively, depending on how the animals dispersed themselves among the two troughs.

All test sessions were video recorded using a camera (GoPro HERO 9, USA) placed centrally 4 m above the two troughs. Videos were assessed by two independent observers using instantaneous scan sampling at 10 second intervals, for a total of 132±0.3 scans per session, or 660±0.72 scans per animal (in five sessions, only 131 scans per animal were recorded due to the camera being switched off prematurely so that the last 10 second scan was not recorded). A heifer was recorded either as feeding from a trough if its muzzle was over the outer edge and inside the trough, or as standing/moving (i.e., muzzle away from the trough). For each behavior, the focal animals' location in the zones of the test arena (see image in S1 Fig) was recorded to assess heifers' social contact during feeding. A heifer was scored as 'feeding alone' if no other heifer was present within ≤ 3 m of the same trough. Heifers were scored as 'feeding together' if at least one companion was also feeding or standing/moving within ≤ 3 m of the same trough. The heifer moving/standing nearby was scored as 'waiting'. A heifer moving/standing alone ≤ 3 m from a trough was assessed as 'waiting alone'. Animals standing/moving at > 3 m distance from the troughs were recorded as 'away' from the troughs. We counted the number of displacements received per individual (i.e., a heifer's head moved out of the trough (stopped feeding) after being pushed or head-butted by another animal; measured as a proxy of competition level [39]). The number of switches an animal performed between troughs was calculated based on the location (≤ 3 m distance to the low or high trough) of the animal at each scan interval. Inter-observer reliability (2 raters) for feeding/not feeding, the feeding side (high or low) and displacements received was excellent (Cohen's kappa = 0.95).

## Home pen observations

Social rank influences cattle's preference for social proximity while feeding [19,20]. We assessed the social rank of each heifer within the group using home pen behaviors video recorded (WV-CW504SP, Panasonic, Japan) for three consecutive 24-h periods (as per [40]). Social hierarchies in cattle are resource-dependent [39]; we recorded successful displacements from the feed bunk for comparison with displacements in the distribution test. We continuously recorded successful agonistic interactions at the feed bunk with outcomes defined as the complete withdrawal (head moving out of the head gate) of the 'reactor' (animals were identified based on their unique coat patterns) following a head butt from the 'actor' [41,42]. One experienced observer assessed all group 1 videos and trained observer 2, who recorded the same videos (inter-observer reliability based on dominance score (see statistical analysis description), icc = 0.67). The second observer assessed videos for groups 2 and 3 (intra-observer reliability, Group 2 icc = 0.68, Group 3 icc = 0.77).

## Statistical analysis

Data analyses were performed in R (version 4.3.2., [43]). To identify personality measures associated with sociability and other traits, we subjected the behaviors observed in the novel arena, novel object and runway tests to principal component analysis (PCA), a statistical procedure commonly applied to assess animal personalities (e.g., [12,34,44]). However,

Kaiser-Meyer-Olkin test showed a mean sampling adequacy (MSA) of 0.36, indicating that our data were unsuitable for PCA [45]. We thus proceeded to test the consistency of outcome measures across tests and test repetition using Spearman rank correlations ($r_s$) for all complete cases (first test repeat n = 33 (two jumped out of the test arena during the novel arena test, and one was unwilling to enter the holding pen for the runway test), second repeat n = 36).

Our trade-off test design differed from previous studies in cattle in that the distance between companions and the maximum distance to obtain feed was much shorter (~22 m in our test compared to 130 m in [24]). Moreover, we also introduced a level of visual separation that likely influenced animals' willingness to move away from peers. We therefore first had to identify which measures are most likely to reflect sociability by assessing the effect of treatment (social versus alone) on outcome measures. We used generalized linear mixed-effect models (GLMM, glmer() from the lme4 package, [46]) to model binary response variables (i.e., animal left or stayed in zone 1, animal reached or did not reach bucket 4) with treatment, treatment order, test repeat and trade-off test trial as fixed effects and subject as a random effect. Continuous measures (latency to leave zone 1, reach bucket 4, return to zone 1 (log-transformed for better model fit)) were modelled in the same way using linear mixed models (LMM, using lmer() from the lme4 package, [46]). To quantify individuals' variation in willingness to trade off social proximity for food, we calculated a trade-off score (latency to leave zone 1 tested in alone condition minus latency to leave in social condition) for each individual that left zone 1 in both conditions for the first time. Animals that did not leave zone 1 on one or both trade-off test trials within five minutes were excluded from the trade-off score calculation. We did this instead of assigning maximum latencies to animals not leaving zone 1, because we could not rule out that these animals remained near bucket 1 because they did not understand that more grain was available in the other buckets, i.e., their response did not reflect a motivational trade-off. We then tested the relationship between the trade-off score (Box-Cox transformed for better model fit) and outcome measures commonly used to assess sociability (e.g., number of vocalizations, runway latency; reviewed in [47]) selected *post hoc* from the social isolation test results (i.e., behaviors found to be consistent across time and tests) using LMM with PT trial as fixed and subject as a random factor.

All animals fed from the buckets in ascending order (except one which was excluded from the analyses that moved from bucket 1–3, then 4, then 2) and all grain available in the buckets was consumed during testing except on one occasion when the animal only consumed about 50% of the grain in bucket 2; this animal was excluded from the analyses. We explored the effect of bucket position, treatment, treatment order, test repeat and trade-off test trial on feeding duration using LMM. We included assessment of feeding times from bucket position and treatment, anticipating that the feeding rate might increase with increasing social stress [48]. Hence, we explored this measure as another potential indicator of sociability.

Heifers' behavior in the distribution test was described as time spent feeding in Ideal Free Distribution, time feeding alone/together or waiting near a feeding conspecific, expressed as the percentage of total test time. We tested the relationship between the dominance index derived from home pen observations and displacements in the distribution test (log-transformed) using a linear model to validate that the individuals' dominance ranks observed in both social situations were significantly associated. Dominance status of individuals within the home pen was calculated with the formula for dominance index proposed by [49]:

$$\left( \frac{N \ cows \ displaced \ by \ focal \ animal}{(N \ cows \ displaced \ by \ focal \ animal + N \ cows \ displacing \ focal \ animal)} \right) * 100$$

Finally, to explore the relationship between sociability measures observed in the presence and absence of social partners, we used LMM to test the effect of trade-off score and personality (i.e., sociability measures derived in social isolation) on individuals' time spent feeding alone in the distribution test (dependent variable). Subgroup (three heifers) nested within group (group 1–3) was used as a random effect. The relationship between the trade-off score and the number of switches between troughs in the distribution test was modelled in the same way to test whether animals more willing to trade off social contact for food (reflected in a higher trade-off score) would also move more between feeding sides.

For all models, model selection was determined using anova() to compare model fit and *p*-values. Individual factors were extracted using the Anova() type II sums of squares function (car package [50]). Model fit was confirmed using the DHARma package [51] to check residual distribution. Least squares means and standard errors were extracted using emm() from the emmeans package [52]. Post hoc comparisons were performed using Tukey-Kramer correction for multiple testing (pairs() from emmeans package). Where appropriate, we report the slope ± standard error (SE) and adjusted or conditional $R^2$. For all statistical tests, *p*-values <0.05 are reported as significant and values between 0.05 and 0.1 as tendencies.

The data set and R code for the statistical analysis of all social motivation measures are available in a public Borealis Dataverse repository at the following link: https://doi.org/10.5683/SP3/AGBUKH.

## Results

### Sociability measured in the absence of conspecifics

Within the novel arena test, only standing ($r_s = 0.66$, $p < 0.001$), vocalization ($r_s = 0.33$, $p = 0.05$) and exploration ($r_s = 0.55$, $p < 0.001$) behaviors were consistent across the two personality test repeats, tested 14 days apart. Likewise, standing ($r_s = 0.4$, $p = 0.02$), vocalization ($r_s = 0.49$, $p = 0.003$), and exploration behaviors ($r_s = 0.41$, $p = 0.02$) recorded in the novel object test were stable across time. Across the novel arena and novel object test, only standing and vocalization behaviors were also positively correlated when tested the first time (standing $r_s = 0.41$, $p = 0.01$, vocalization $r_s = 0.78$, $p < 0.001$) and when repeated 14 days later (standing $r_s = 0.42$, $p = 0.01$, vocalization $r_s = 0.70$, $p < 0.001$). Exploration behaviors were not consistent across novel arena and novel object tests in the two repeats (first repeat: $r_s = 0.26$, $p = 0.13$; second repeat: $r_s = 0.32$, $p = 0.07$).

Runway test vocalizations emitted by the test animals were stable across time ($r_s = 0.51$, $p = 0.002$), but calls emitted in the runway test were not correlated with vocalizations in the NAT or NOT (for both test days, all $p > 0.05$). Runway test latencies showed temporal stability ($r_s = 0.38$, $p = 0.03$) over the two repeats.

### Social rank assessment

For each of the three home pen observation days, we recorded on average $202 \pm 7.21$ successful displacements from the feed bunk in Group 1; $188 \pm 25.86$ events in Group 2; and $278 \pm 29.46$ displacements in Group 3. The number of feed bunk displacements recorded per day (range 165–310) exceeded 120, the minimum number of events required for the appropriate statistical analysis required for a group of 12 animals [53]. From these interactions, we calculated individuals' dominance index relative to their group, which was used as an alternative explanatory factor for behaviors observed in the distribution and trade-off tests.

### Trade-off test results

Animals' willingness to leave or stay in zone 1 was not affected by treatment (social *versus* alone condition, $X^2_1 = 0.08$, $p = 0.77$). There was no effect of treatment order ($X^2_1 = 0.07$, $p = 0.79$), or trial repeat within the same test day ($X^2_1 = 0.09$, $p = 0.76$). However, we noted an effect of test repeat ($X^2_1 = 13.07$, $p < 0.001$) with heifers being less likely to leave when tested the first time (left = 33, stayed = 37; results from both trade-off trials combined) than when tested 14 days later (left = 52, stayed = 20). Five heifers did not leave zone 1 in any of the test trials.

Of the animals leaving zone 1, treatment ($X^2_1 = 4.68$, $p = 0.03$) and test repeat ($X^2_1 = 15.99$, $p < 0.001$) affected latency to leave, with no interaction between these factors ($X^2_1 = 0.04$, $p = 0.82$). Based on these results, we concluded that latency to leave zone 1 was a sensitive measure of social motivation. We calculated a trade-off score (latency to leave while alone minus latency to leave in social condition) for all animals that left zone 1 in both trade-off trials for the first time (n = 25), and assessed the relationship between this measure and sociability measures derived from other social contexts (see below). Social dominance index was not associated with the trade-off score ($-1.40 \pm 0.94$, $R^2 = 0.04$, $p = 0.15$).

When tested alone, heifers were faster to leave zone 1 than when tested in the social condition (t=−2.172, p=0.03) and also faster to leave when tested in the second test repeat compared to when tested the first time (t=3.8, p<0.001). Latency to leave was not affected by trade-off trial ($X^2_1$=2.41, p=0.12) or treatment order ($X^2_1$=2.46, p=0.11).

The likelihood to reach bucket 4 was not predicted by treatment ($X^2_1$=0.11, p=0.73), but was by test repeat ($X^2_1$=6.19, p=0.01) and trade-off test trial within the same test repeat ($X^2_1$=4.41, p=0.03). More animals reached bucket 4 in the second test repeat (n=22) than in the first test repeat (n=10, z=−2.47, p=0.01), and the second trade-off trial (n=21) compared to the first trade-off trial (n=17, z=−2.1, p=0.03). Latency to reach bucket 4 was predicted by test repeat (−21.06±8.89, $R^2$=0.59, p=0.02), with shorter latencies in the second test repeat than when tested the first time (first test repeat: 108.28±35.57 s; second test repeat: 103.92±40.36 s, t=2.34, p=0.02). None of the other factors affected latency to reach bucket 4 (treatment: $X^2_1$=1.7, p=0.19; treatment order: $X^2_1$=0.16, p=0.68; trade-off test trial: $X^2_1$=0.006, p=0.93). Hence, the willingness to approach the bucket further away was not deemed a sensitive measure for sociability and therefore dropped from further analysis.

Feeding duration was influenced by bucket position ($X^2_3$=136.72, p<0.001), with shorter feeding durations at buckets 3 and 4 compared to buckets 1 and 2 (all p<0.05). Heifers fed longer at bucket 1 compared to bucket 2 (t=7.68, p<0.001), but feeding duration was not different at buckets 3 and 4 (t=0.52, p=0.95). Feeding durations were shorter at the second test repeat 14 days later ($X^2_1$=7.22, p=0.007) but longer when animals were tested for the second time on the same day (trade-off test trial effect, $X^2_1$=6.29, p=0.01). There was only a trend in treatment effect on feeding times ($X^2_1$=3.21, p=0.07), with feeding times tending to be longer in the social than in the alone condition. This measure was dropped from further analysis, given the lack of effect of the treatment condition.

Latency to return from bucket 4 was predicted by treatment ($X^2_1$=12.97, p<0.001), but contrary to our predictions, animals were slower to return to zone 1 when tested in the social condition compared to when tested alone (social: 139.57±120.94 s, alone: 74.00±63.49 s, t=−3.57, p=0.001). Given the inconsistencies in expected and observed results, we explored the effect of the number of vocalizations emitted by the companion animals during testing after noticing that companions vocalized twice as much when in visual contact with the test animal compared to when they were not (22.85±16.3 calls vs. 14.0±11.5 calls). However, companion vocalizations did not influence return latencies of the test animal (see S6 Appendix for details of statistics and results). Animals were faster to return in the second test repeat than in the first (first test repeat: 138.56±90.77 s; second test repeat: 93.96±105.23 s, t=3.94, p<0.001). Treatment order ($X^2_1$=3.51, p=0.15) and trade-off test trial ($X^2_1$=3.18, p=0.13) did not affect return latency. We dropped this outcome variable from further analysis, given that the reason why animals took longer to reestablish proximity to conspecifics was unclear.

### Distribution test

Across the 12 subgroups, animals dispersed according to the Ideal Free Distribution (IFD) (with one heifer feeding alone from the low side and two feeding together from the high side) 24.94±24.87% of the total test time. There were differences in dispersal among subgroups and test days. Some subgroups never split up (0% in IFD), while others followed IFD for up to 82.44% of the time. In the tests where heifers deviated from IFD, all three animals fed together from the same side (high side: 20.05±18.84%, low side: 16.76±14.42% of the time) most of the time and occupied the two troughs non-ideally (i.e., one feeding on the high, two on the low side) 8.14±12.62% of the time. In only 1.07±1.63% of the time was there no heifer feeding, while 29.01±8.48% of the time, one of the three heifers was not feeding (i.e., either waiting near a trough or standing/moving in the back of the test area).

At the individual level, time spent feeding alone ranged from 0 to 56.60%. Feeding alone, heifers did so for longer from the low side (64.42±35.24%) than from the high side (33.90±34.49%). Averaged across all five test sessions, individuals switched between feeding sites 19.15±11.66 times (range: 1–37) and received 7.69±8.66 displacements (range: 0–44).

A higher dominance index was negatively associated with a lower number of displacement heifers received during the distribution test (−5.44±2.40, $R^2$=0.10, $p$=0.03), confirming that these measures of social dominance were related. However, the dominance index was not associated with the number of times animals switched between troughs (−0.02±0.13, $R^2$=−0.02, $p$=0.83). Moreover, feeding alone was not predicted by the number of displacements received (−0.06±0.27, $R^2$=0.001, $p$=0.83) or dominance index (0.24±0.15, $R^2$=0.06, $p$=0.13).

## Relationship between social motivation measures

Comparing the social-feed trade-off willingness with measures derived from the social isolation tests, the trade-off score was not predicted by novel arena vocalizations (−0.58±1.36, $R^2$=0.01, $p$=0.67), runway vocalizations (0.65±2.74, $R^2$=−0.01, $p$=0.81), or runway latency (0.36±0.48, $R^2$=−0.02, $p$=0.46).

Likewise, time spent feeding alone was not predicted by the vocalizations emitted in social isolation (novel arena vocalizations: −0.13±0.21, $R^2$=0.01, $p$=0.54; runway vocalizations: −0.04±0.42, $R^2$=0.03, $p$=0.92). There was a trend in longer runway latency predicting more time feeding alone (0.06±0.04, $R^2$=0.08, $p$=0.09).

Comparing sociability measures within different social contexts, we found that time spent feeding alone in the distribution test was not predicted by individuals' trade-off score (−0.03±0.04, $R^2$=0.03, $p$=0.52, see Fig 3).

## Discussion

There is increasing recognition that considering individual variations in animals' preferences and motivations is key for ensuring a high level of animal welfare. Individual preference for social proximity (i.e., sociability) differs among individual cattle [3]. Social context may affect the expression of inter-individual variations in personality traits, such as sociability, potentially leading to variations in behavior when alone versus in groups [30,32]. Conversely, personality traits are expected to be stable across different contexts [14].

We found that sociability inferred from behaviors observed in standardized social isolation tests (i.e., novel arena, novel object, and runway tests) was not associated with social motivation associated with social-feed trade-off willingness (i.e., propensity to leave conspecific proximity in exchange for grain). Our results agree with previous findings reporting that sociability evaluated via social-feed trade-offs does not correspond with cattle's responses measured during social separation (i.e., novel object test, restraint test; [33]). The lack of consistency may be due to the behavior displayed in the

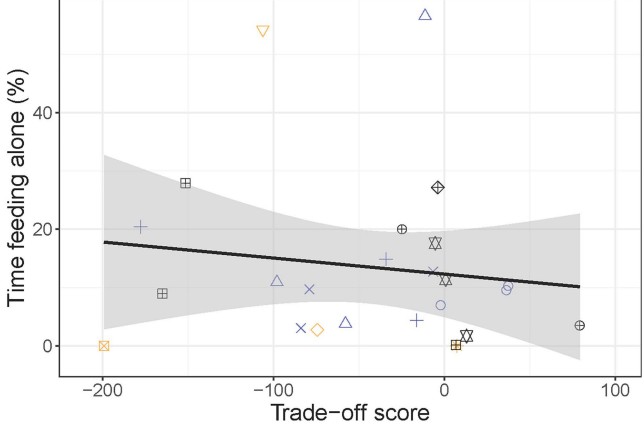

**Fig 3. Relationship between individuals' time spent feeding alone in the distribution test and social-feed trade-off score.** Both measures were anticipated to reflect willingness to trade off social contact for feed, but no relationship between measures was observed (−0.03±0.04, $R^2$=0.03, $p$=0.52).

social isolation and trade-off tests reflecting different aspects of social motivation. While forced social separation induces fear and stress [15,16], social-feed trade-off tests reflect a propensity to voluntarily distance from conspecifics in exchange for food. That is, we compared test scenarios that tap into opposite motivational systems, with food reward likely activating the reward-seeking system and social isolation inducing avoidance responses, guiding different behavioral responses in animals (e.g., [54]). Moreover, in situations where animals can behave proactively (i.e., have agency over their actions) their responses to their environment may differ from reactions under conditions where actions do not produce the desired results [55]. Exposing dairy cattle to the same type of personality test (novel object test) alone or in a group, Foris et al. [30] reported a positive association between responses expressed in both conditions. This result suggests that when exposed to the same situations (novelty), behavioral variations among individuals may be stable across social and asocial contexts.

Here, we aimed to understand the relationship among measures of sociability across different social situations. We observed heifers' distribution among two feeding sites in the distribution test, intending to observe social feeding behavior in an ecologically meaningful context where animals have the opportunity to disperse proportionally to resource availability. Studies in free-ranging herbivores (sheep) have shown that animals may deviate from IFD due to motivational conflicts between group cohesion *versus* access to forage [22]. We hypothesised that heifers' propensity to feed alone (i.e., trading social contact for higher feed reward) would be associated with lower levels of sociability as measured in the other tests. However, we failed to find any relationship between social feeding behavior and sociability estimated from social isolation tests, perhaps due to mismatches in motivation systems as discussed above. Moreover, differences in the magnitude of behavioral expressions induced by the social isolation and distribution test may have resulted in discrepancies in sociability measures. Behavioral responses may be enhanced or suppressed in non-social and social situations, leading to behaviors observed in one condition not being associated with another [32]. For instance, in the absence of group members, certain types of behaviors may be upregulated in the attempt to re-establish group proximity (e.g., increase in vocalization and activity [15]), but the same behavior may be reduced in the presence of conspecifics. We found that vocal responses were expressed frequently during social isolation but rarely observed in the tests with conspecifics present, suggesting that vocal responses are context-dependent. Vocal responses may indicate motivation to regain social contact with conspecifics [15,16], but may not be comparable with alternative sociability measures such as social spacing behavior. Interestingly, other personality traits (e.g., exploration, fearfulness, activity) inferred from social isolation test have been reported to correlate with home pen behaviors in dairy cattle (e.g., more exploratory and more active calves consumed more solid feed; [56,57]).

Individual variations in behaviors might be affected at different levels when animals are tested in non-social and social conditions [32]. For example, fish resumed foraging more rapidly after a simulated predator attack when tested in groups than they did when tested alone, and this difference was more notable for more active compared to less active fish [58]. Hence, the social isolation tests might have affected our test animals in different ways, potentially in interaction with other intrinsic factors such as fearfulness or coping style [18].

Comparing temporal and across-test stability of measures derived from social isolation tests, very few outcome measures (namely, vocalization, standing, exploration, and runway latency) showed within-individual consistency. Our findings add to the evidence that test-retest and inter-test reliability are not always achieved with social isolation tests (reviewed in [7,59]). One reason for this could be different rates of habituation among individuals. The social isolation tests were repeated within the same test arena (with slight modifications, see methods), and so were the tests with social contact. This might have led to some heifers being more affected by periods of social isolation, or possibly induced frustration caused by a mismatch in anticipated and actual testing events (i.e., being put into a test arena without food or conspecifics after having been exposed to both on previous occasions), possibly leading to the inconsistency in behavior.

Because of the low repeatability of outcome measures in the social isolation tests, we were unable to apply Principal Component Analysis (PCA). PCA is often used to identify personality traits by condensing several correlated behavioral

measures into principal components [45]. Our results echo concerns about the suitability of the social isolation tests to assess specific personality traits in gregarious animals [7] and, adding fuel to the larger debate on the value of standardized testing of animal behavior [60].

Intending to compare sociability in a more species-specific context, we evaluated heifers' social motivation in the presence of conspecifics. Previous research proposed that the motivational conflict in social-feed trade-off is an influential factor in how animals distribute themselves among feeding sites [22]. To our knowledge, no other study has tested IFD in indoor-housed dairy cattle. Although animals showed considerable differences in their propensity to move away from conspecifics in exchange for feed, time feeding alone in the distribution test was not associated with the individuals' trade-off score. While the social-feed trade-off test allowed each individual to autonomously move away from peers to obtain more grain, the time each heifer spent feeding alone or together in the distribution was not independent of the behavioral decision made by the companion heifers. In free-ranging sheep, individuals' with low sociability (i.e., those that are comfortable to move away from peers on their own) might still be observed near animals with higher sociability due to the decision of the latter [22], suggesting that an individuals' position within a group might not always reflect its preferred level of social proximity. Hence, the lack of a relationship between both measures might be attributed to the distribution test being less representative of individuals' voluntary trade-off willingness.

Our test groups often deviated from IFD in a manner consistent with high levels of social motivation. Unfortunately, we are unable to disentangle whether heifers perceived social proximity while feeding as beneficial or whether they merely 'tolerated' sharing their feeding space. The benefits of the competition for and monopolization of a resource are context-dependent. For example, when individuals are hungry and food availability is limited, they have an interest in competing with one another to gain access to the resource. However, when satiated, displaying aggression has little benefit and there is some evidence that adult cattle would passively share food (i.e., show little agonistic behaviors around the resource, [61]). In our study, the relative social rank among heifers did not affect their distribution among the two troughs, contrary to previous observations that subordinate cattle avoid feeding near more dominant animals [19,20]. Increased competition to access highly valued feed can flatten the dominance hierarchy in adult cows [21]. Therefore, individuals assessed with a lower dominance index under less competitive conditions (home pen) might have been more persistent in sharing the same feed bunk space with more dominant conspecifics when offered grain than when offered lower-value feed. This idea is supported by the findings that displacements from the feed troughs did not result in animals switching to the alternative trough or spending more time alone. Future refinements of similar group feeding tests should include grouping heifers intentionally according to their social ranking or providing less desired feed rewards (e.g., hay) when comparing social spacing as a potential measure of social motivation with other sociability measures. Finally, we encourage continued research to explore the conditions under which the IFD model is applicable to confined social animals.

## Conclusion

Two important conclusions can be drawn from our findings: 1) social motivation appears to be context-dependent and may not be generalizable across situations with and without social contact, and 2) intrinsic characteristics (e.g., fearfulness) and external factors (e.g., level of novelty in social isolation tests, whether observed behavior is independent from that of social partners or not) may influence how behaviors manifest within and across social contexts. We suggest that sociability should not be understood as a single 'trait', but rather as a behavioral consequence resulting from the combined effects of internal and external factors at the time of evaluation.

Our study continues the line of inquiry that emphasizes the need to consider animals as individuals that differ in their behavioral responses to their environment, and highlights the complexity of finding ways to scientifically assess intrinsic attributes modulating animal behavior.

## Supporting information

**S1 Appendix. Sample size estimation.**
(DOCX)

**S2 Appendix. Randomization of animals and treatment groups.**
(DOCX)

**S3 Appendix. Inter-observer reliability testing for novel arena and novel object test measures.**
(DOCX)

**S4 Appendix. Distribution test habituation.**
(DOCX)

**S5 Appendix. Grain amount determination for the distribution test.**
(DOCX)

**S6 Appendix. Companion vocalization analysis.**
(DOCX)

**S1 Fig. Image of the distribution test arena.**
(TIF)

**S1 Table. Ethogram of behaviors assessed in the social isolation tests.**
(DOCX)

## Acknowledgments

We thank the staff of The University of British Columbia (UBC) Dairy Research and Education Center, UBC Animal Welfare Program students and visiting students for their support with this study. We particularly thank Inez Sham, Shirley Yang, João Pedro Donadio, Laura Field, Karim Amziane, Diane Buffiere, Maya Thompson, Lucy Macdonell and Christina Doelling from the Animal Welfare Program for their help with animal handling, conducting the experiments and data collection.

## Author contributions

**Conceptualization:** Sarah Kappel, Emeline Nogues, Daniel M. Weary, Marina A.G. von Keyserlingk.

**Data curation:** Sarah Kappel, Emeline Nogues, Daniel M. Weary.

**Formal analysis:** Sarah Kappel, Emeline Nogues, Daniel M. Weary.

**Funding acquisition:** Marina A.G. von Keyserlingk.

**Investigation:** Sarah Kappel, Emeline Nogues, Marina A.G. von Keyserlingk.

**Methodology:** Sarah Kappel, Emeline Nogues, Daniel M. Weary, Marina A.G. von Keyserlingk.

**Project administration:** Daniel M. Weary, Marina A.G. von Keyserlingk.

**Resources:** Marina A.G. von Keyserlingk.

**Software:** Marina A.G. von Keyserlingk.

**Supervision:** Daniel M. Weary, Marina A.G. von Keyserlingk.

**Validation:** Sarah Kappel, Emeline Nogues.

**Visualization:** Sarah Kappel, Emeline Nogues, Marina A.G. von Keyserlingk.

**Writing – original draft:** Sarah Kappel, Emeline Nogues.

**Writing – review & editing:** Daniel M. Weary, Marina A.G. von Keyserlingk.

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
