## [Decision Letter · Decision Letter 0]

20 Aug 2025

Dear Dr. von Keyserlingk,

Thank you for submitting your manuscript to PLOS ONE. After careful consideration, we feel that it has merit but does not fully meet PLOS ONE’s publication criteria as it currently stands. Therefore, we invite you to submit a revised version of the manuscript that addresses the points raised during the review process.

**Please see below my comments and the reviewers' suggestions.**

We look forward to receiving your revised manuscript.

Kind regards,

Luis Alonso Villalobos

Academic Editor

PLOS ONE

Journal Requirements:

[NSERC Discovery]. 

3. Please expand the acronym “NSERC” (as indicated in your financial disclosure) so that it states the name of your funders in full.

4. Please upload a copy of your Supporting Information files, to which you refer in your manuscript. If the Supporting Information files are no longer to be included as part of the submission please remove all reference to it within the text.

Additional Editor Comments:

Dear Dr Marina A.G. von Keyserlingk

Thanks for your patience while we completed the amount of reviews for your manuscript. Please find attached the comments from the reviewers and proceed to edit and submit a revised version of the manuscript.

I appreciate the order and great flow of the ideas throughout the whole document. Don't hesitate to contact me in case of any question regarding the submission.

Reviewers' comments:

Reviewer's Responses to Questions

**Comments to the Author**

1. Is the manuscript technically sound, and do the data support the conclusions?

Reviewer #1: Yes

Reviewer #2: Yes

2. Has the statistical analysis been performed appropriately and rigorously?

Reviewer #1: Yes

Reviewer #2: Yes

3. Have the authors made all data underlying the findings in their manuscript fully available?

Reviewer #1: Yes

Reviewer #2: Yes

4. Is the manuscript presented in an intelligible fashion and written in standard English?

Reviewer #1: Yes

Reviewer #2: Yes

Reviewer #1: Line 18 - Area should be arena

Line 50 - There is a possibility that sociability may also be separation distress. Jack Panksepp calls this PANIC

Panksepp, J. (2011) The basic emotional circuits in mammalian brains: Do animals have affective lives? Neuroscience and Biobehavioral Reviews, 35(2011), 1791-1804

Line 54 - I agree that social isolation may reflect more than on personality trait.

Line 117 - State the breed of the heifers

Discussion - You need to think about the differences between birds and sheep and your dairy heifers. Both birds (27) and grazing sheep (22) have greater freedom to move about in the space where they forage. It is also possible that foraging birds and grazing sheep had greater hunger motivation than your very well fed heifers. The birds and sheep are also not in a situation where they could be eating from the same feed. Could the birds and sheep living in a more extensive environment had an effect on the results?

This reference may be helpful. It shows that two scores that are often being used for measuring cattle temperament may be measuring different traits.

Bruno, K. et al. (2016) Relationship between quantitative measures of temperament and other observed behaviors in growing cattle, Applied Animal Behavioral Science, 199:56-66.

Reviewer #2: This is an excellent manuscript that has presented a complex range of results from varying behavioural tests and the relationships between the test outcomes.

I struggled to find ways to improve the text.

Lines 39-40 reads a bit like a repeat of the previous text.

Line 65: should this be 'in exchange for feed;'?

Line 195: delete double use of 'contact'

Lines 326-328: can you state how many animals this equates to for not leaving zone one.

**Do you want your identity to be public for this peer review?** For information about this choice, including consent withdrawal, please see our Privacy Policy

Reviewer #1: **Yes: ** Temple Grandin

Reviewer #2: No

---

## [Author Response · Author response to Decision Letter 1]

8 Sep 2025

Dear reviewers,

Thank you very much for taking the time to review our manuscript and provide comments to help us improve our article. We address specific comments below:

Reviewer #1:

Line 18 - Area should be arena

AU; Corrected.

Line 50 - There is a possibility that sociability may also be separation distress. Jack Panksepp calls this PANIC

Panksepp, J. (2011) The basic emotional circuits in mammalian brains: Do animals have affective lives? Neuroscience and Biobehavioral Reviews, 35(2011), 1791-1804

AU: Thank you. We agree with the reviewer and have added the suggested reference and the text “or as a signal of separation distress” in L 51.

Line 54 - I agree that social isolation may reflect more than on personality trait.

AU: Thank you.

Line 117 - State the breed of the heifers

AU: Added in L 118.

Discussion - You need to think about the differences between birds and sheep and your dairy heifers. Both birds (27) and grazing sheep (22) have greater freedom to move about in the space where they forage. It is also possible that foraging birds and grazing sheep had greater hunger motivation than your very well fed heifers. The birds and sheep are also not in a situation where they could be eating from the same feed. Could the birds and sheep living in a more extensive environment had an effect on the results?

AU: We agree that free-ranging animals might differ in their willingness to disperse from peers in exchange for food, especially if feed is available less consistently. We specified that the discussed example of sheep behaviour was done in free-ranging animals by added “free-ranging” before sheep in L 570.

We added “Finally, we encourage continued research to explore the conditions under which the IFD model applies to confined social animals” in L 596 – 597 to clarify that the suitability of IFD as a framework for measuring social motivation in confined animals needs further investigation.

This reference may be helpful. It shows that two scores that are often being used for measuring cattle temperament may be measuring different traits.

Bruno, K. et al. (2016) Relationship between quantitative measures of temperament and other observed behaviors in growing cattle, Applied Animal Behavioral Science, 199:56-66.

AU: Thank you for your suggestion. After reviewing the article, we have decided not to include this reference, as we regard personality and temperament as distinct concepts. Given the breadth of information already presented to the reader, we believe that introducing an additional concept may lead to unnecessary complexity.

Reviewer #2:

This is an excellent manuscript that has presented a complex range of results from varying behavioural tests and the relationships between the test outcomes.

I struggled to find ways to improve the text.

AU:Thank you.

Lines 39-40 reads a bit like a repeat of the previous text.

AU:Corrected in L 39. We replaced the repetition of “reunited with” with “return to its group”.

Line 65: should this be 'in exchange for feed;'?

AU:Corrected in L 66.

Line 195: delete double use of 'contact'

AU;Corrected in L 196.

Lines 326-328: can you state how many animals this equates to for not leaving zone one.

AU;We state the number of animals leaving and not leaving zone in the results section, L 401 and 402.

---

## [Decision Letter · Decision Letter 1]

23 Sep 2025

Exploring dairy heifers' consistency in social motivation in the absence or presence of conspecifics

PONE-D-25-30150R1

Dear Dr. von Keyserlingk,

We’re pleased to inform you that your manuscript has been judged scientifically suitable for publication and will be formally accepted for publication once it meets all outstanding technical requirements.

Kind regards,

Luis Alonso Villalobos

Academic Editor

PLOS ONE

Additional Editor Comments (optional):

Dear author

Thanks for your patience while we completed the review of the revised version of the manuscript.

The reviewers found the changes satisfactory and I am glad to tell you that your manuscript has been accepted for publication.

Reviewers' comments:

Reviewer's Responses to Questions

**Comments to the Author**

Reviewer #1: All comments have been addressed

Reviewer #2: All comments have been addressed

2. Is the manuscript technically sound, and do the data support the conclusions?

Reviewer #1: Yes

Reviewer #2: Yes

3. Has the statistical analysis been performed appropriately and rigorously?

Reviewer #1: Yes

Reviewer #2: Yes

4. Have the authors made all data underlying the findings in their manuscript fully available?

Reviewer #1: Yes

Reviewer #2: Yes

5. Is the manuscript presented in an intelligible fashion and written in standard English?

Reviewer #1: Yes

Reviewer #2: Yes

Reviewer #1: The authors have addressed my comments. They changed area to arena and added the Panksepp, J. (2011) reference for separation stress. They also clarified their comparison between the behavior of birds, sheep, and fed heifers. I agree with the author's decision to not use the Bruno et al 2016 reference. Please accept the paper for publication.

Reviewer #2: (No Response)

**Do you want your identity to be public for this peer review?** For information about this choice, including consent withdrawal, please see our Privacy Policy

Reviewer #1: **Yes: ** Dr. Temple Grandin

Reviewer #2: No

---

## [Editor Report · Acceptance letter]

PONE-D-25-30150R1

PLOS ONE

Dear Dr. von Keyserlingk,

I'm pleased to inform you that your manuscript has been deemed suitable for publication in PLOS ONE. Congratulations! Your manuscript is now being handed over to our production team.

Kind regards,

on behalf of

Dr. Luis Alonso Villalobos

Academic Editor

PLOS ONE